



# A Predictive Model for Salt Nanoparticle Formation Using Heterodimer Stability Calculations

Sabrina Chee[1], Kelley Barsanti[2], James N. Smith[1], and Nanna Myllys[1,3]

[1]Department of Chemistry, University of California, Irvine
[2]Department of Chemical & Environmental Engineering, University of California, Riverside
[3]Department of Chemistry, University of Jyväskylä

**Correspondence:** Nanna Myllys (nanna.myllys@helsinki.fi), James N. Smith (jimsmith@uci.edu)

**Abstract.** Acid–base clusters and stable salt formation are critical drivers of new particle formation events in the atmosphere. In this study, we explore the relationship between $J_{1.5}$, the theoretically predicted formation rate of clusters larger than 4 acid and 4 base molecules, and acid–base heterodimer stability, a property that is relatively easy to calculate using computational methods. Heterodimer stability as a function of gas-phase acidity, aqueous-phase acidity, heterodimer proton transference, vapor pressure, dipole moment, and polarizability were explored for the salts comprised of sulfuric acid, methanesulfonic acid, and nitric acid with nine bases. The best predictor of heterodimer stability was found to be gas-phase acidity. The relationship between heterodimer stability and $J_{1.5}$ was analyzed for sulfuric acid salts over a range of monomer concentrations from $10^5$ to $10^9$ molec cm$^{-3}$ and temperatures from 248 to 348 K. Heterodimer concentration was calculated from heterodimer stability and yielded an expression for predicting $J_{1.5}$ for any salt, given approximately equal acid and base monomer concentrations and knowledge of monomer concentration and temperature. This parameterization was tested for the sulfuric acid–ammonia system by comparing the predicted values to experimental data and was found to be accurate within 2 orders of magnitude. We show that one can create a simple parameterization that incorporates the dependence on temperature and monomer concentration on $J_{1.5}$ by defining a new term that we call the normalized heterodimer concentration, $\Phi$. A plot of $J_{1.5}$ vs. $\Phi$ collapses to a single monotonic curve for all weak salts of sulfuric acid, and can be used to accurately estimate $J_{1.5}$ in atmospheric models.

## 1 Introduction

Atmospheric aerosol particles represent the largest uncertainty in our understanding of global climate through their participation in cloud formation and the absorption and scattering of radiation (Kerminen et al., 2005; Kuang et al., 2009; Lohmann and Feichter, 2004; Merikanto et al., 2009; Spracklen et al., 2008). In particular, particle formation by nucleation is still not well understood and is difficult to represent in models (Kerminen et al., 2018). One of the dominant nucleation pathways is through salt formation, where the formation of a cluster is stabilized by the interactions between acid and base molecules, which enhances particle formation (Ball et al., 1999; Kirkby et al., 2011; Kürten et al., 2016; Nadykto and Yu, 2007; Nadykto et al., 2015; Wang et al., 2018). This nucleation pathway is particularly dominant in urban environments, where anthropogenic sources for acidic and basic gases are abundant (Ge et al., 2011; Kirkby et al., 2011; Qiu and Zhang, 2013; Weber et al., 1996; Wang et al., 2020). Although sulfuric acid ($H_2SO_4$, sa) is most commonly associated with atmospheric nucleation (Ball et al.,





1999; Bzdek et al., 2012; Kirkby et al., 2011; Angelino et al., 2001; Weber et al., 1995), nitric acid (HNO$_3$, na) and methane-sulfonic acid (CH$_3$SO$_3$H, msa) have been also observed to be participants and may also play important roles in the initial stages of cluster growth (Afpel et al., 1979; Barsanti et al., 2009; Mäkelä et al., 2001; Smith et al., 2004, 2008; Weber et al., 1995), the latter of which we shall refer to henceforth as new particle formation (NPF).

5     Ammonia is the most abundant base in the atmosphere and its reaction with sulfuric acid has been well studied (Bzdek et al., 2010; Glasoe et al., 2015; Weber et al., 1996). Alkylamines have also garnered attention due to their high basicity and demonstrated ability to enhance NPF more than ammonia, despite their lower atmospheric abundance (Kurtén et al., 2008; Smith et al., 2010; Temelso et al., 2018; Waller et al., 2019; Kreinbihl et al., 2020).

    Recently, computational efforts have focused on accurately representing the formation and growth of acid–base clusters (Smith 10 et al., 2021). Myllys et al. (2016a) investigated the accuracy of the domain local pair natural orbital coupled cluster (DLPNO–CCSD(T)) method, and found that it allows for the modeling of up to 10 molecules in a cluster, which had been previously not feasible with other highly accurate methods. The DLPNO–CCSD(T)/aug-cc-pVTZ//$\omega$B97X-D/6-31++G** level of theory has become increasingly popular for modeling atmospheric processes such as cluster formation of sulfuric acid with ammonia, methylamine, dimethylamine, trimethylamine, guanidine, monoethanolamine, trimethylamine N-oxide, and a variety of 15 diamines (Myllys et al., 2016a; Ma et al., 2019; Xie et al., 2017; Myllys et al., 2020, 2018; Elm et al., 2016, 2017).

    This large variety in systems studied has yielded insights into the factors that determine cluster formation and growth. Generally, the enhancing efficiency of the base on heterodimer stability and NPF is known to correlate with base strength, which has been attributed to a more favorable proton transfer and the formation of essentially nonvolatile ionic salts and has been shown to be generally true for the most abundant bases in the atmosphere: ammonia, methylamine, dimethylamine, 20 and trimethylamine (Almeida et al., 2013; Elm, 2017; Myllys et al., 2019b; Barsanti et al., 2009; Shen et al., 2020; Han et al., 2020). For many studies that observe both cluster and nanoparticle formation and growth, p$K_a$ has been often used as the metric for basicity. However, since p$K_a$ is, by definition, an aqueous measure of acidity, applying it to cluster and nanoparticle-sized systems does not take into account the drastically different environment. Indeed, in the study by Xie et al. (2017), monoethanolamine (p$K_a$ = 9.5) enhanced NPF more than methylamine (p$K_a$ = 10.6), despite methylamine being the 25 stronger base according to their p$K_a$ values (Haynes, 2014). In that study, the lack of a base strength trend was attributed to the additional hydrogen bonding sites provided by the -OH group on monoethanolamine. In addition, we have recently studied the modelled formation rates of sulfuric acid and trimethylamine-N-oxide (tmao), guanidine, or dimethylamine, where tmao, despite its lower basicity (p$K_a$ = 4.7) to both guanidine (p$K_a$ = 13.6) and dimethylamine (p$K_a$ = 10.7), had similar formation rates to guanidine, which were much higher than those of dimethylamine (Myllys et al., 2020; Haynes, 2014). In these studies, 30 p$K_a$ was insufficient to predict NPF enhancement.

    In this study, we aim to use these computational methods to identify what molecular properties predict heterodimer stability, or more specifically the Gibbs free energy of formation of the heterodimer ($\Delta G_{\text{heterodimer}}$), and in turn, formulate a model to predict NPF rate. We specifically investigate the use of p$K_a$ in comparison to gas-phase acidity measures to predict proton transfer in the heterodimer as well as heterodimer stability. In addition, we examine if base vapor pressure has any correlation 35 to heterodimer stability, as sulfuric acid is often cited to participate in NPF because of its low volatility and condensation onto





clusters (Weber et al., 1996; Ball et al., 1999; Sipilä et al., 2010). Finally, we also calculate the dipole moment and polarizability of the studied base molecules to see if, in the absence of ions, they have any predictive capability of heterodimer stability. These observations extend to salts of sa, msa, and na with nine bases: ammonia (amm), methylamine (ma), dimethylamine (dma), trimethylamine (tma), trimethylamine N-oxide (tmao), guanidine (gua), monoethanolamine (mea), putrescine (put) and

piperazine (pz) (Table 1).

In addition to these molecular properties, we further explore the relationship between heterodimer stability and NPF rate for sa salts. The goal of this work is to develop computationally efficient approaches for calculating NPF rate that can be applied to models that estimate the impacts of NPF on climate and air quality. We represent NPF rate as $J_{1.5}$, the rate at which a cluster larger than 4 acid and 4 base molecules is formed. A cluster of this size can range in diameter from 1 to 1.5 nm, depending

on the constituent acid and base. We analyze the relationship between heterodimer stability and the theoretically predicted $J_{1.5}$ for sulfuric acid salts over a range of monomer concentrations from $10^5$ to $10^9$ molec cm$^{-3}$ and temperatures from 248 to 348 K. The concentration of heterodimers was calculated from heterodimer stability, temperature, and monomer concentrations for the case where acid and base monomer concentrations are approximately equal. This results in a parametrization for $J_{1.5}$ as a function of heterodimer concentration that can be applied to any acid–base system. These results were compared to $J_{1.7}$

rates measured at the CLOUD (Cosmics Leaving OUtdoor Droplets) chamber for sa–amm salts. We note that the relationship between $J_{1.5}$ and heterodimer concentration is not unique, but depends on both temperature and monomer concentration. However if the dependent variable is redefined as a term that we call the "normalized heterodimer concentration," or $\Phi$, then a simple monotonic relationship develops that can be used to predict $J_{1.5}$ for all weak salts of sulfuric acid, a system that is of great interest in the atmosphere. We believe that this approach is generalizable to any acid–base system, allowing accurate

predictions of NPF rates over a wide range of monomer concentration, temperature, and ambient pressure.

## 2   Computational methods

Two-component acid–base particle formation was studied by making systematic changes in temperature and concentration to understand the effects of simulation conditions and acid/base molecular properties on $J_{1.5}$. Correlations of $J_{1.5}$ with different molecular properties provided insight into the critical factors of cluster formation. Properties listed in Table 2 were examined

as possible variables that may have a role in stabilizing clusters and enhancing particle formation.

### 2.1   Cluster thermodynamics

In order to simulate cluster formation and growth, one must calculate accurate structures and thermochemical properties of neutral sa–base clusters up to the cluster size of four sa and four base molecules (4sa4base). Thermochemistry of clusters containing amm, dma, gua and tmao were taken from our previous studies (Myllys et al., 2018, 2019b, 2020). Thermochemistry

of clusters with mea, put and pz were taken from a database (Elm, 2019), collected from original publications of Xie et al. (2017), Elm et al. (2017) and Ma et al. (2019). Available structures with ma and tma were taken from Olenius et al. (2017) and, to be consistent with the level of theory used, structures were optimized and frequencies calculated at the $\omega$B97X-D/6-





**Table 1.** Acid and base compounds in this study. Abbreviations are as follows: ammonia (amm), methylamine (ma), dimethylamine (dma), trimethylamine (tma), trimethylamine N-oxide (tmao), guanidine (gua), monoethanolamine (mea), piperazine (pz), putrescine (put), sulfuric acid (sa), methanesulfonic acid (msa), and nitric acid (na).

| amm | ma | dma | tma | tmao |
|---|---|---|---|---|
| $NH_3$ | $H_2N-CH_3$ | | | |

| gua | mea | pz | put |
|---|---|---|---|

| sa | msa | na |
|---|---|---|

**Table 2.** Experimental and calculated properties examined in this study.

| Property | Source |
|---|---|
| Gas phase acidity (GA) | calculated this work |
| Difference between GA of an acid HA and a conjugate acid of a base $BH^+$ ($\Delta$GA) | calculated this work |
| Aqueous phase acidity ($pK_a$) | from Haynes (2014) |
| Difference between $pK_a$ of HA and $BH^+$ ($\Delta pK_a$) | from Haynes (2014) |
| Vapor pressure | from literature[a] |
| Electrochemical properties: dipole moment and polarizability | calculated this work |
| Solvation free energy difference between a base B and its protonated form $BH^+$ ($\Delta\Delta$SOL) | calculated this work |
| Heterodimer stability ($\Delta G_{\text{heterodimer}}$, free energy of a complex having one acid and one base) | calculated this work |
| Remaining H-bond donors on base molecule in heterodimer | inferred |
| Proton transfer in heterodimer | inferred |

[a]Stull (1947); Aston et al. (1937, 1939); Swift and Hochanadel (1945); Matthews et al. (1950); EPISUITE v 4.11





31++G** level and electronic energies corrected at the DLPNO–CCSD(T)/aug-cc-pVTZ level with TightPNO, TightSCF, and GRID4 keywords. In addition, for the missing structures, we performed a configurational sampling as explained in Kubečka et al. (2019). For the lowest free energy clusters, Gibbs free binding energies were calculated at the DLPNO–CCSD(T)/aug-cc-pVTZ//ωB97X-D/6-31++G** level of theory (Riplinger and Neese, 2013; Riplinger et al., 2013; Myllys et al., 2016a, b).

In addition of a full data set for sa–base clusters, we studied heterodimers of na and msa with above-mentioned nine bases. The same quantum chemical methods were used as in sa–base calculations. In order to detect whether proton transfer was occurring in the heterodimer, the Molden program (Schaftenaar and Noordik, 2000) was used to visualize the global minimum structure. Gas-phase basicity and proton affinity values were computed using the same level of theory. Gaussian 16 RevA.03 (Frisch et al., 2016) was used to optimize geometries and calculate vibrational frequencies and Orca version 4.2.1 (Neese, 2012) was

used for single point energy corrections.

## 2.2    Particle formation simulations

Theoretical methods allow us to perform particle formation simulations at any conditions. This means that very low or high temperatures and vapor concentrations can be used to estimate $J_{1.5}$. While some values in this range might not be directly "atmospherically relevant," these calculations can lead to a deeper understanding of the non-linear behavior of nucleation as

a function of vapor concentrations and/or temperature. It is also possible to study cluster formation of different compounds under identical conditions because there are no instrumental limitations or measurement biases.

The calculated thermodynamic data sets for sa–base clusters were used as input in Atmospheric Cluster Dynamics Code (ACDC), which detailed theory is explained in McGrath et al. (2012). Briefly, the ACDC model simulated particle formation by solving the cluster distribution considering collision, evaporation and removal processes. The model calculated the rate

constants for each process among the population of clusters and vapor molecules and solved the discrete general dynamic equations for each cluster type. We have performed $J_{1.5}$ simulations at temperatures of 248–348 K using sa and base vapor concentrations of [acid]=[base]=$10^5$–$10^9$ cm$^{-3}$. Simulated $J_{1.5}$ values are given in the supporting information (SI). Simulations were performed for neutral clustering pathways at dry conditions due to computational (quantum chemical) restrictions.

## 3    Results and discussion

### 3.1    Heterodimer stability results

In the cluster formation process, the changes in enthalpy ($\Delta H$) and entropy ($\Delta S$) are always negative because hydrogen bond formation is an exothermic process in which the degrees of freedom are decreasing when isolated molecules become one entity. Gibbs free energy is calculated from $\Delta H$ and $\Delta S$ as a function of temperature by

$$\Delta G = \Delta H - T\Delta S \tag{1}$$

where $\Delta G$ decreases as temperature decreases. Lower $\Delta G_{\text{heterodimer}}$ values correspond to more stable heterodimers. However, while a negative $\Delta G_{\text{heterodimer}}$ value indicates a spontaneous reaction in solution at standard conditions, heterodimer formation





in the gas phase under atmospheric conditions also depends on the acid and base vapor concentrations. Table 3 presents enthalpies, entropies and Gibbs free energies of sa–base heterodimer formation at 298 K and corresponding tables for msa and na are given in the SI. From these data, heterodimer stability can be calculated at other temperatures readily for all 27 salts studied here. Our calculated $\Delta G_{\text{heterodimer}}$ value for sa–amm indicates a less stable heterodimer than the sa–amines

heterodimers, which is consistent with numerous other studies (Kurtén et al., 2008; Nadykto et al., 2011; Leverentz et al., 2013; Kupiainen et al., 2012). Ma and mea are the weakest heterodimer stabilizers among the amines; dma, tma and pz are stronger and form approximately equally stable heterodimers. Of these nine bases, the most stable heterodimers are formed with tmao, gua, and put.

**Table 3.** Calculated enthalpy ($\Delta H_{\text{heterodimer}}$ in kcal/mol), entropy ($\Delta S_{\text{heterodimer}}$ in cal/(mol·K)) and Gibbs free energy ($\Delta G_{\text{heterodimer}}$ in kcal/mol) for sa–base heterodimer formation at 298 K.

| BASE | $\Delta H_{\text{heterodimer}}$ | $\Delta S_{\text{heterodimer}}$ | $\Delta G_{\text{heterodimer}}$ |
|------|------|------|------|
| amm  | −15.1 | −29.2 | −6.4 |
| ma   | −18.2 | −33.6 | −8.2 |
| dma  | −22.2 | −30.3 | −13.2 |
| tma  | −23.6 | −35.2 | −13.1 |
| tmao | −32.2 | −34.9 | −21.8 |
| gua  | −29.4 | −30.4 | −20.3 |
| mea  | −21.8 | −38.2 | −10.4 |
| put  | −28.9 | −44.8 | −15.6 |
| pz   | −22.8 | −33.3 | −12.9 |

The molecular structures of sa–base heterodimers are presented in Fig. 1 and for msa and na heterodimers in the SI. Amm

is the only base which is unable to accept a proton from sa in the heterodimer structure; the heterodimer is held together via one hydrogen bond between amm and sa. All other base compounds accept a proton from sa and form an ion pair with the deprotonated sa, bisulfate. Protonated tma and tmao form only one hydrogen bond with bisulfate, whereas the other bases form two hydrogen bonds. In the sa–put heterodimer, put also forms an intramolecular hydrogen bond via its protonated and non-protonated amino groups.

## 3.2   Molecular properties that affect heterodimer stability ($\Delta G_{\text{heterodimer}}$)

### 3.2.1   Evaluation of gas-phase versus aqueous-phase acidity

Figure 2 shows that gas phase and aqueous phase basicity values do not trend the same amongst the nine bases. For $NR_3$ compounds, where R is either H or $CH_3$, the gas-phase monomer basicities directly follow the number of substitutions as amm < ma < dma < tma. This means that when removing a proton from isolated gas-phase $BH^+$ compound, the Gibbs free reaction

energy has the largest value in the case of tma. That is because the methyl groups stabilize cation formation by distributing



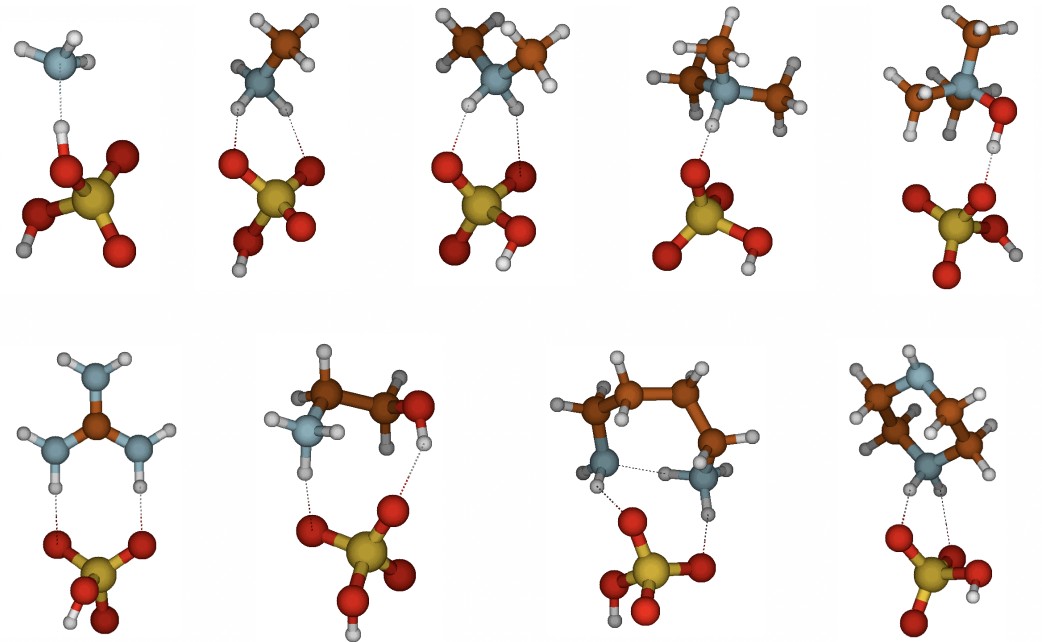

**Figure 1.** Heterodimers of sa with amm, ma, dma, tma, tmao, gua, mea, put and pz, respectively.

the charge. In the aqueous-phase ($pK_a$), however, the basicities have an different order: amm < tma < ma < dma. This means that dma has the largest proportion of protonated base cations in water solution. Dma has two methyl groups that facilitate protonation, and H-bond formation with water molecules provides additional stabilization. In the case of tma, the hydration is very limited due to the steric hindrance of three methyl groups, and thus, tma has lower aqueous-phase basicity than dma and

ma. Because the basicity order of amines in the gas-phase directly follows the substitution order, the anomalous inversion of basicities in aqueous-phase can be attributed to the stabilization effect of surrounding solvent molecules (Seybold and Shields, 2015).

In the gas phase, the strongest bases are, in decreasing order, put, tmao and gua, whereas in the aqueous phase the order is gua, put and dma. Gua is a very strong base both in gas and aqueous phase because its cationic form has six $\pi$-electrons that

are delocalized over the Y-shaped plane. This $D_{3h}$-symmetric structure of guanidinium makes it extraordinarily stable. Tmao is very strong base in the gas phase because of its zwitterionic bond, where oxygen has a negative charge that strongly attracts $H^+$. In the aqueous phase, polar solvent molecules are capable of stabilizing the zwitterionic bond in tmao, thus tmao is the weakest base in the water solution. The reason why put is the strongest base in the gas phase is related to the change of its configuration between neutral and cationic forms. The neutral form of put is linear, but the cation is cyclic as the protonated and

deprotonated amino groups are hydrogen bonded to each other as shown in Fig. 3. The Gibbs free energy difference between cyclic global minimum configuration and lowest acyclic local minimum configuration is 14.6 kcal/mol, which is the additional stabilization caused by the H-bond in gas phase. The gas basicity of put calculated based on the acyclic form would be 215.2





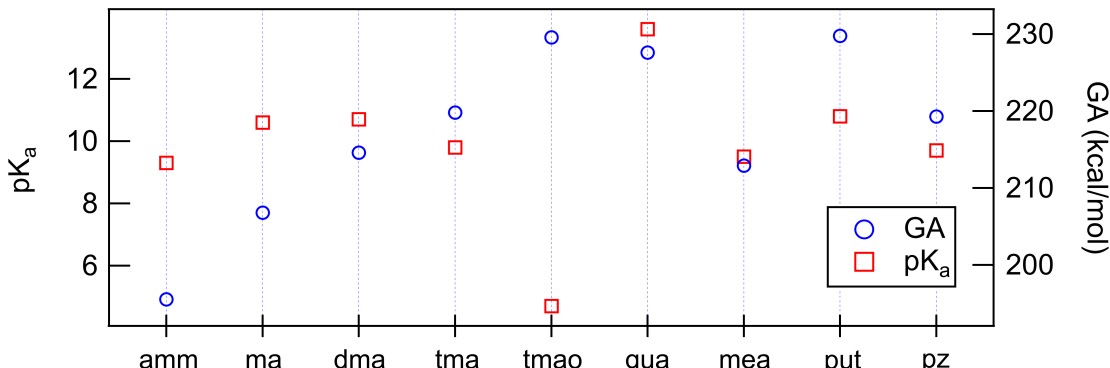

**Figure 2.** Calculated GA vs literature $pK_a$ values from Haynes (2014).

kcal/mol, which is very close to that of dma — and interestingly the $pK_a$ values of dma and put are very close to each other. This could indicate that protonated put is in aqueous phase mainly in its acyclic form and is stabilized by H-bonds with water molecules in the same manner as dma.

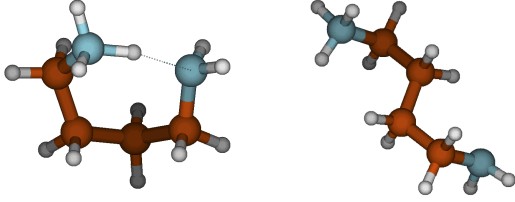

**Figure 3.** Cyclic and acyclic configurations of protonated put. The lowest energy acyclic structure is 14.6 kcal/mol higher in free energy than the cyclic, global minimum structure.

    As put and pz are diamines, they can accept two protons and form $baseH_2^{2+}$ cations. The PA and GA values for the second
5  protonation reaction are significantly smaller than for the first protonation reaction: for put 130.6 and 125.2 kcal/mol and for
pz 121.0 and 113.3 kcal/mol, respectively. While the PA and GA values can be measured for the first protonation reaction for
each base, there was no experimental data found for the second protonation reaction. Experimental PA and GA values from
Hunter and Lias (1998) are given in the SI and good agreement with our calculated values is shown. PA, GA and $pK_a$ values
are listed for sa, msa and na in the SI.
10    Because heterodimer stability has been shown to be a good proxy for $J_{1.5}$, we have plotted the correlation between $\Delta G_{\text{heterodimer}}$
and $\Delta GA$ and $\Delta pK_a$ to probe the hypothesis that acid and base strength predict the formation of the heterodimer (Figure 4).
Here $\Delta GA$ is defined to be the difference between the GA of the acid and the GA of the protonated base. And similarly the
$\Delta pK_a$ value is defined as the difference between the $pK_a$ of the acid and the $pK_a$ of the protonated base. All $pK_a$ values

were taken from literature as bulk aqueous phase dissociation constants, whereas GA values were calculated for this study. By definition, the larger the $\Delta$GA, the less favorable the acid–base reaction is in the gas phase. Similarly, the more positive the $\Delta pK_a$, the less favorable the acid–base reaction is in the bulk aqueous phase.

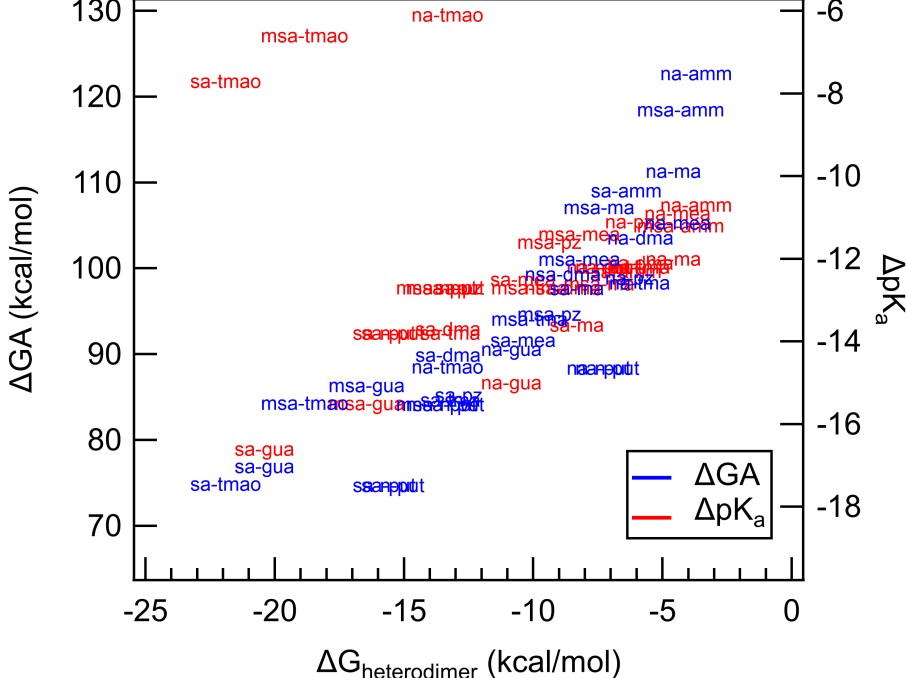

**Figure 4.** Calculated $\Delta$GA and $\Delta pK_a$ plotted against $\Delta G_{\text{heterodimer}}$. Each data point represents an acid–base pair between either sa, na, or msa with either amm, ma, dma, tma, tmao, gua, mea, put, or pz. Blue text represent $\Delta$GA values, while red text represent $\Delta pK_a$ values. Text markers are centered over the data point.

Over the observed $\Delta$GA, as $\Delta$GA increases, the less stable the heterodimer. The story is similar for $\Delta pK_a$: as $\Delta pK_a$ increases, the heterodimer becomes less stable. However, for $\Delta pK_a$, tmao salts seem to deviate drastically from the trend. Indeed, this is most likely because tmao is more able to be stabilized by water molecules in the bulk aqueous phase and its proton exchange in the gas phase is not well represented by $pK_a$. Otherwise, the trend of $\Delta pK_a$ matches up well with that of $\Delta$GA. These results demonstrate that acid and base strength have a clear relationship with $\Delta G_{\text{heterodimer}}$, and that $\Delta$GA can be used in parameterizations of $\Delta G_{\text{heterodimer}}$. $\Delta$GA is even less computationally intensive than $\Delta G_{\text{heterodimer}}$ because it only models the removal of a proton from the original molecule in comparison to modeling the interactions of two molecules. In addition, GA values can be calculated for an array of acids and bases to get $\Delta$GA for a larger combination of acids and bases rather than modelling $\Delta G_{\text{heterodimer}}$ for each acid–base pair. For example, in this study, 3 acids and 9 bases were studied: to calculate $\Delta$GA for all combinations, only 12 reactions need to be simulated; in contrast, $\Delta G_{\text{heterodimer}}$ would need to be calculated for each of the 27 salts. Because the GA values calculated here agree well with those experimentally determined





in Hunter and Lias (1998), this modeling approach may be a simpler, more consistent method to predict GA values for yet-unstudied bases, including those that are atmospherically relevant.

Figure 5 illustrates how $\Delta$GA is a better predictor of proton transfer in the gas phase than $\Delta$p$K_a$. In general, acid–base pairs with $\Delta$GA of 103 kcal/mol or below undergo proton transfer, and thus $\Delta$GA provides a threshold for cluster formation. This

is consistent with the stronger trends between heterodimer stability and GA than heterodimer stability and p$K_a$, the latter of which was affected by the solubilities of the acids and bases, which is not relevant to cluster formation and growth in the gas phase.

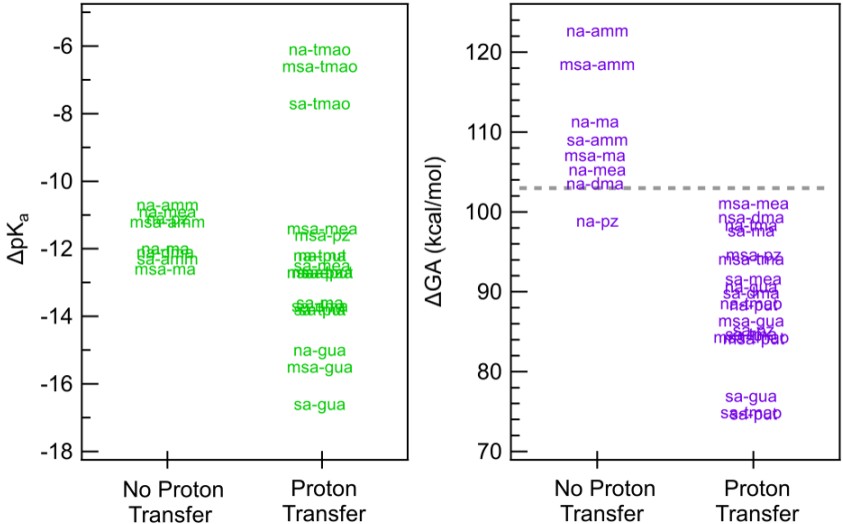

**Figure 5.** $\Delta$GA and $\Delta$p$K_a$ values separated based on whether the heterodimer structure exhibits proton transfer. The grey dashed line on the $\Delta$GA graph at 103 kcal/mol shows the cutoff point for proton transfer.

Interestingly, the na–pz salt is an anomaly in the cutoff for $\Delta$GA in predicting proton transfer, with $\Delta$GA value of 98.9 kcal/mol, yet there is no proton transfer in the global minimum structures of heterodimer. However, there exists a local mini-

10 mum structure in which proton transfer occurs that is only 1.8 kcal/mol higher in free energy than the global minimum. Figure 6 shows that in the proton transferred form of the na–pz pair, the second H-bond formation, which is needed to stabilize the anion–cation pair, is unfavorable because of the induced ring strain. Generally, na is less likely to form two H-bonds with a base than sa or msa as the angle of O-N-O is 120° whereas the O-S-O angle in sa and msa are 109°, and therefore the ring strain would be high in na salts (with an exception for gua as shown in the SI). Overall, heterodimer proton transfer only occurs in

clusters with a $\Delta$GA smaller than 103 kcal/mol (na–put) with the exception of na–pz. In general, this strengthens the idea that $\Delta$GA is a better estimate of gas-phase reactivity than $\Delta$p$K_a$ and emphasizes the importance of using thermodynamic constants that accurately represent the systems being studied.

$\Delta$GA and $\Delta$p$K_a$ values can and should be used in lab settings to gauge the likelihood of nucleation. For example, numerous studies, including those in our own lab, show that oxalic acid does not form particles with any of the methylated amines (ma,





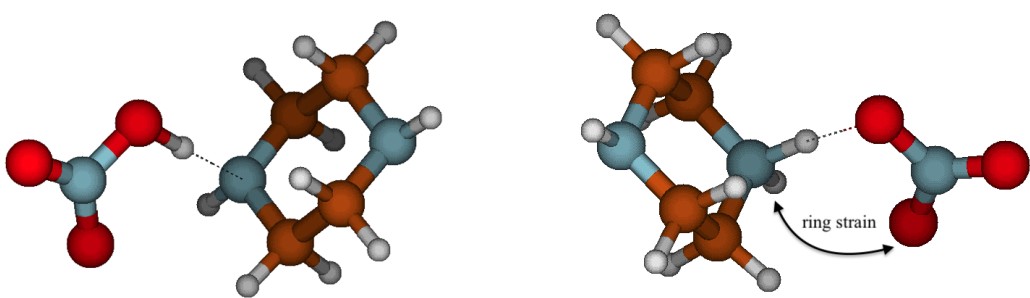

**Figure 6.** Deprotonated (left) and protonated (right) conformers of pz for the na–pz salt showing the ring strain necessary to form another intermolecular hydrogen bond.

dma, tma) in a two-component system at 298 K (Arquero et al., 2017). The most negative $\Delta pK_a$ value for these oxalic acid salts is $-9.45$, which is more positive than any of the systems studied here. Considering that na–amm does not form particles at room temperature even at high concentrations, its $\Delta pK_a$ value of $-10.7$, or its $\Delta GA$ value of 122.65 kcal/mol, could be used as a benchmark for predicting particle formation at room temperature. This cutoff is dependent on both temperature and the

concentrations of precursor acid and base and should be viewed as a qualitative means for predicting NPF at room temperature. A more accurate means of estimating NPF rates that accounts for both temperature and precursor concentration is presented in Section 3.3.1.

### 3.2.2  Factors that do not affect heterodimer stability

Figure 7 shows the relationship between base vapor pressure and heterodimer stability ($\Delta G_{\text{heterodimer}}$), which is plotted to
explore the hypothesis that the volatility of the base, which is typically much higher than that of the accompanying acid, is a limiting factor that drives NPF. The lack of correlation suggests that acid–base reactive uptake, leading to salt formation, is the dominant mechanism and that volatility of the constituent acid and base plays a relatively minor role in heterodimer stability. It is important to emphasize that this lack of correlation between vapor pressure and heterodimer stability is only observable because the bases have different structural properties. Otherwise, if only amm, ma, dma, and tma were studied,
then trends for vapor pressure and heterodimer stability would follow the trend of the more volatile base making a less stable heterodimer, which is untrue. Since the most well-studied bases in the atmosphere are amm, ma, dma, and tma, due to their relative abundance and contribution to NPF, it may be tempting to make conclusions on base behavior in NPF based solely on those four bases. However, these correlations — or lack thereof — highlight the importance of a wider breadth of study for us to better understand how bases behave in the atmosphere. This disappearance of a trend as more bases are included applies to
the dipole moment and polarizability of the base as well (see SI). However, it is worth noting that while base vapor pressure does not affect heterodimer stability, it may have a larger role in determining particle composition as particles grow to a size that represents bulk systems (Lawler et al., 2016; Chen and Finlayson-Pitts, 2017; Myllys et al., 2020; Chee et al., 2019).





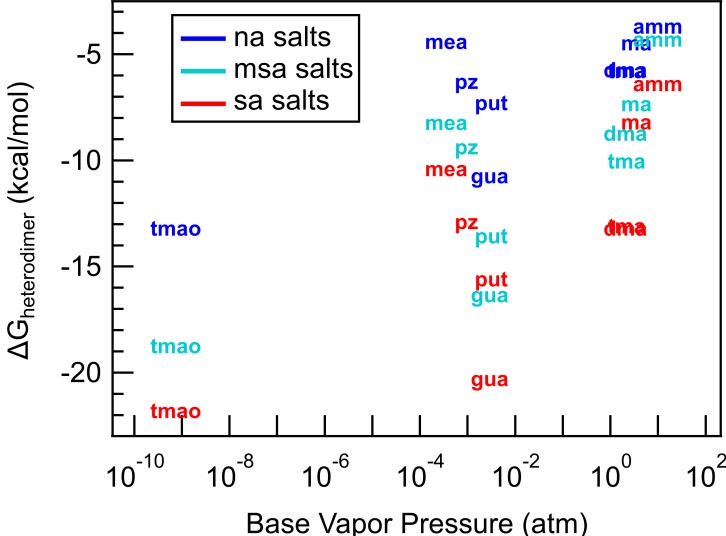

**Figure 7.** Base vapor pressure plotted against $\Delta G_{\text{heterodimer}}$ for sa, msa, and na salts.

### 3.3 Heterodimer stability versus $J_{1.5}$

The stabilities of a heterodimer and other small clusters are known to affect the ability of a cluster to grow to a large aerosol particle (Almeida et al., 2013; Elm, 2017; Olenius et al., 2013) We now correlate $\Delta G_{\text{heterodimer}}$ with calculated $J_{1.5}$ for all nine bases with sa at varying conditions to observe the change in new particle formation rate over the temperature range of 248–348 K (Figure 8a), and acid and base monomer concentrations from $10^{5}$–$10^{9}$ molec cm$^{-3}$ (Figure 8b). For reference, a $J$ of 0.1 cm$^{-3}$s$^{-1}$ is also indicated, which can be viewed as a lower limit for observed atmospheric $J_{1.5}$ (Kerminen et al., 2018). We emphasize that these some of the concentrations and temperatures might not be very common in the atmosphere. However, through these systematic changes in temperature and concentrations, we are able to gain insight into the predictors of cluster formation and growth.

Previously, theoretically calculated $J_{1.5}$ for reactions of sa with dma or amm have been shown to be a good approximation for experimentally determined NPF rates observed at the CLOUD chamber (Myllys et al., 2019b). As Figure 8a shows, $J_{1.5}$ follow a lognormal relationship with $\Delta G_{\text{heterodimer}}$. This makes sense in that, for the most stable heterodimers like salts of tmao and gua, $J_{1.5}$ approaches the kinetic limit and simply cannot form any faster. However, as heterodimer stability decreases, the evaporation of a heterodimer occurs faster than its collision with vapor molecules or other clusters, which results in a reduction in $J_{1.5}$. In contrast, when temperature is held constant and base concentration is varied (Figure 8b), the lognormal relationship remains the same across all cases and only the maximum $J_{1.5}$ is shifted until the kinetic limit is reached. The changing relationship between $\Delta G_{\text{heterodimer}}$ and $J_{1.5}$ with varying temperature can be attributed to the change in the thermodynamics of the reaction, while the shift in NPF rate with respect to $\Delta G_{\text{heterodimer}}$ with varying concentration can be attributed to the





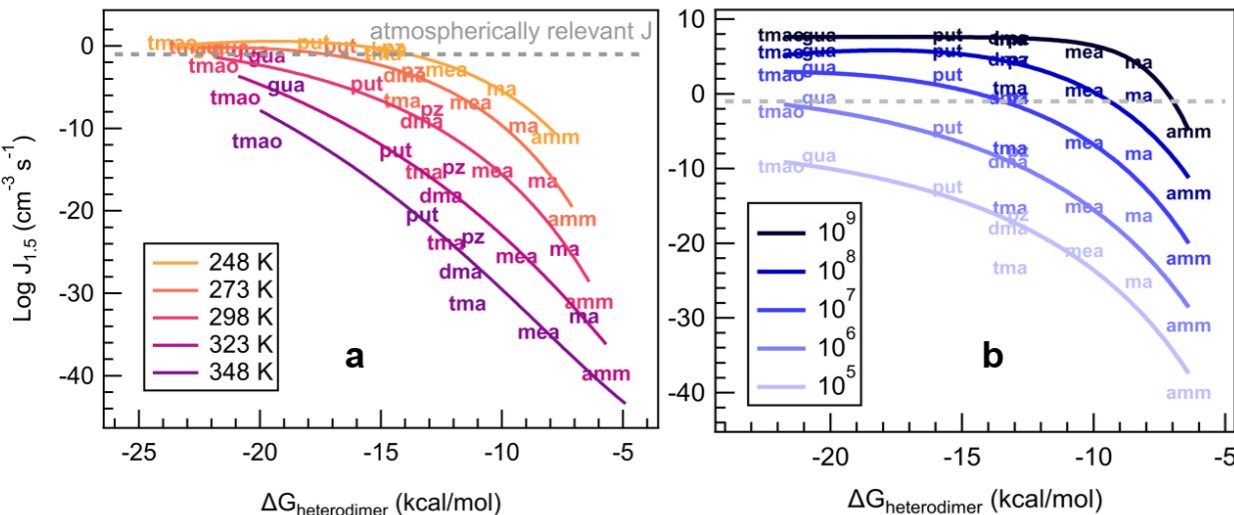

**Figure 8.** Heterodimer stability ($\Delta G_{\text{heterodimer}}$) plotted against NPF rate ($J_{\text{theory}}$) in varied conditions. a) Vapor concentrations are constant: [acid]=[base]=$10^6$ molec cm$^{-3}$ at varying temperature: $T$=248, 273, 298, 323, and 348 K. b) Temperature is constant: $T$=298 K at varying vapor concentrations: [acid]=[base]=$10^5$, $10^6$, $10^7$, $10^8$, and $10^9$ molec cm$^{-3}$. Text markers are centered over the data point.

relatively higher number of collisions in a shorter period of time. This behavior matches the relationship of $J$ with temperature and concentration found in classical nucleation theory (Arstila et al., 1999; Trinkaus, 1983; Vehkamäki et al., 2002):

$$J = Z * p(1,2) * \exp\left[\frac{-(W - W(1/2))}{RT}\right], \tag{2}$$

where $J$ is the nucleation rate, $Z$ is a kinetic pre-factor, $W$ is the work of formation of the critical nucleus, and $p(1,2)$ and 5 $W(1,2)$ are number concentration and cluster formation energy, respectively. Concentration is directly proportional to $J$, whereas temperature contributes from within the exponential expression, which matches the behavior seen in Figure 8.

Interestingly, as temperature increases, this lognormal relationship transitions to linear, with a larger spread of data points around the trendline. Practically, this implies that $\Delta G_{\text{heterodimer}}$ predicts theoretical $J_{1.5}$ well at cold temperatures, but additional factors become more prominent at warmer temperatures. To understand what processes are important for $J_{1.5}$, we scaled the 10 color on each of the bases to the number of hydrogen bond donors (HBD) remaining on the heterodimer after the proton was transferred at two temperatures (Figure 9). The number of remaining HBD was determined by counting the number of polar hydrogens on the base molecule minus the hydrogen donated by sa (if the proton transfer reaction occurred). Although other intermolecular H-bonds exist, those were not subtracted because as the cluster grows, those bonds are broken as the base shifts to accommodate an additional molecule. Sa salts with ma, tmao, put, and gua salts all demonstrate this behavior, where the 15 intermolecular bonds present in the heterodimer for ma, put, and gua are rearranged with each added molecule to the cluster (see SI).



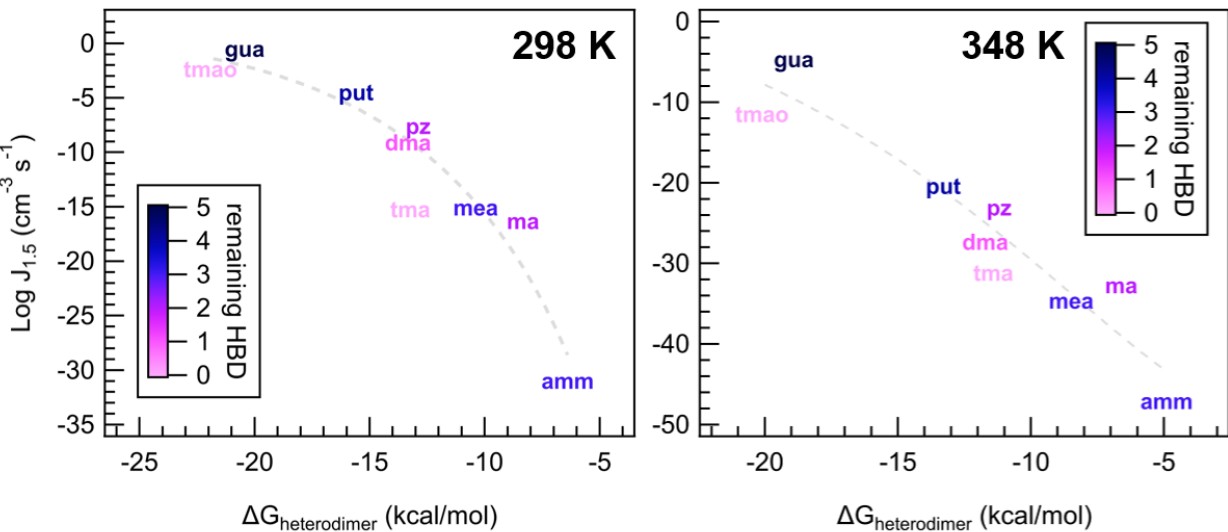

**Figure 9.** Individual data points and trendlines from Figure 8a colored according to the number of remaining hydrogen bond donors (HBD) on the heterodimer. The left is data from the 298 K case, and the right is data from the 348 K case, at [acid]=[base]=$10^6$ molec cm$^{-3}$.

With respect to the lognormal relationship between $J_{1.5}$ and $\Delta G_{\text{heterodimer}}$, tma and tmao, and to a lesser extent, dma, are below the trendline, and they have 0–1 remaining HBD. In contrast, amm, ma, mea, pz, and put have 2–4 remaining HBD and are closest to the trendline. Gua is the only molecule that has 5 remaining HBD, and consistently has a higher NPF rate than the trendline suggests. This behavior can be attributed to cluster growth being slightly dependent on how well the next molecules can "stick" onto the existing cluster, where if there are more remaining HBD on a cluster, it is easier and faster for the cluster to grow. It is interesting that ma has higher NPF rates than the trendline compared to mea, put, and pz despite having either the same or one fewer HBD, but this may be attributed to the bulkiness of the alkyl groups attached to those amines, which may block the remaining HBD from participating in stabilizing the growing cluster.

These findings are notable in that $\Delta G_{\text{heterodimer}}$ trends consistently with $J_{1.5}$ and deviations from these trendlines can be attributed to structural differences in the base, where a base with more HBD available on the heterodimer would have higher predicted NPF rates than the trendline, with the inverse also being true. However, $\Delta G_{\text{heterodimer}}$ varies strongly with temperature and concentration as described above, and as such is not conducive to predicting $J_{1.5}$, which we attempt to remedy in the following two sections.

### 3.3.1 A generalized parameterization to predict $J_{1.5}$

In order to combine simulated particle formation rates at different conditions for all acid–base systems, we calculated the heterodimer concentration, which is a function of $\Delta G_{\text{heterodimer}}$, temperature, and the concentration of the gaseous acid and base monomers. The stability of a heterodimer defines its theoretical maximum concentration at given conditions assuming the





system is at equilibrium. Assuming mass balance for the heterodimer formation reaction leads to the following concentration under equilibrium conditions:

$$[\text{heterodimer}] = \frac{[\text{acid}][\text{base}]}{C_{\text{ref}}} \exp\left(-\frac{\Delta G_{\text{heterodimer}}}{RT}\right). \tag{3}$$

The equilibrium concentration of the heterodimer [heterodimer] is dependent both on the Gibbs free formation energy $\Delta G_{\text{heterodimer}}$

5 (calculated at reference concentration $C_{\text{ref}} = \frac{P_{\text{ref}}}{RT}$, where $P_{\text{ref}}$ is defined as 1 atm and $C_{\text{ref}}$ is in units of molec cm$^{-3}$), and on the monomer concentrations [acid] and [base]. Here we use heterodimer concentration to estimate $J_{1.5}$ under any (atmospherically relevant) temperature or concentration. However, as different acid–base systems form particles via different pathways depending on acid-to-base ratios, the NPF mechanism may change when either the acid or base is in excess. Thus the derivations here are directly applicable at situations when acid and base concentrations are close to equal.

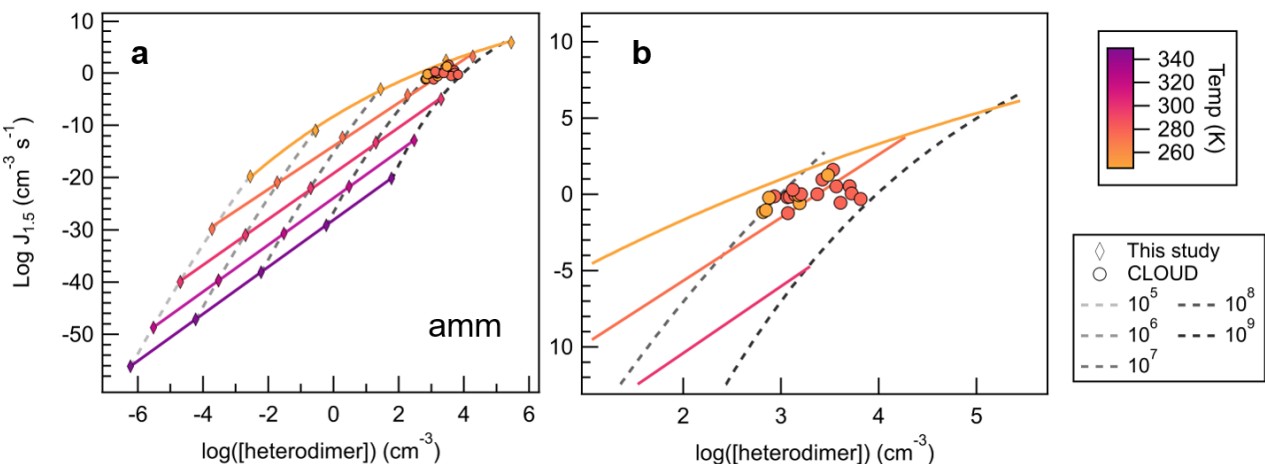

**Figure 10.** Heterodimer concentration plotted against $J_{1.5}$ for sa–amm across 25 computational conditions (filled diamonds) from 248–348 K and monomer concentrations from $10^{5}$–$10^{9}$ cm$^{-3}$, where a) shows the full set of conditions calculated for sa–amm, and b) shows a magnification of how CLOUD data compares to the computational dataset. Colored lines correlate to temperature trendlines that were drawn through all data points calculated at the same temperature. Dashed lines represent data points calculated at the same monomer concentrations. We calculated heterodimer concentrations for CLOUD data whose acid and base concentrations were within 50% of each other according to Equation 3. All CLOUD data points were collected at temperatures of either 248 or 273 K (colored circles corresponding to color scale) and with monomer concentrations between approx. $10^{8}$–$10^{9}$ cm$^{-3}$.

10    Figure 10a shows the temperature and concentration effects on heterodimer concentration for sa–amm salts. As one would expect from Equation 3, as concentration increases, heterodimer concentration increases by two orders of magnitude (as reflected in the [heterodimer] term). However, because temperature affects both the calculation of $\Delta G_{\text{heterodimer}}$ and heterodimer concentration, this relationship is not as simple. In general, as temperature decreases, heterodimer concentration increases. As heterodimer concentration increases and temperature decreases, $J_{1.5}$ also increases, though we begin to see $J_{1.5}$ begin to





saturate at 248 K and $10^9$ cm$^{-3}$. Through the use of heterodimer concentration, we have been able to combine the two factors, temperature and monomer concentration, into one term, where we can now use it to compare (or predict) $J_{1.5}$.

To test the robustness of our calculations, heterodimer concentrations of CLOUD experiments were calculated using Equation 3 and this study's calculated $\Delta G_{\text{heterodimer}}$ values to compare our $J_{1.5}$ calculations to CLOUD's measured J$_{1.7}$ (Kirkby

et al., 2011). Because heterodimer concentration can only be calculated for experiments run at approximately equal acid and base concentrations, all experiments that had more than a 50% difference between monomer concentrations were excluded. Twenty-one measured J$_{1.7}$ values met this criterion and are shown as filled circles in Figure 10. When using the closest temperature trendlines (i.e., CLOUD data measured at 273 K was compared to 278 K model trend) to predict the CLOUD data, the difference between the predicted and measured $J$ were within 2 orders of magnitude. On the other hand, if concentration

trendlines were used to predict $J$ (i.e, CLOUD vapor concentrations were near $10^8$ molec cm$^{-3}$ so the modelled $10^8$ molec cm$^{-3}$ trendline was used), differences of up to 4 orders of magnitude occurred. Trendline equations for sa–amm are shown in the SI, as well as difference plots to show the accuracy of the trendlines as discussed.

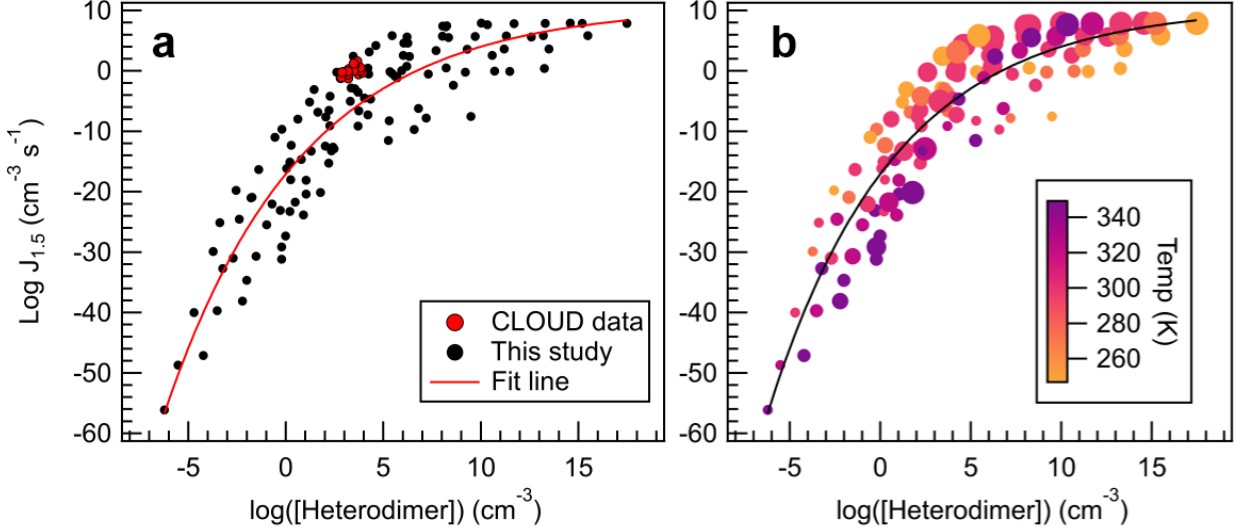

**Figure 11.** Heterodimer concentration plotted against $J_{1.5}$, wherein a) all data are represented with black dots, and b) data points are colored according to temperature and sized to reflect monomer concentrations ($10^5$–$10^9$ cm$^{-3}$). Data were fitted to an exponential function, which can be found in Equation 4.

All data calculated for this study are plotted in Figure 11, which spans 100 K and 5 orders of magnitude in monomer concentrations. Indeed, concentration and temperature effects are minimized compared to the direct comparison between $J_{1.5}$

and $\Delta G_{\text{heterodimer}}$ (Figure 8). Because more $J_{1.5}$ were calculated for amm and gua, data points were left as black points to avoid





complicating the data. Data were fitted to give the following equation:

$$J_{1.5} = 10.688 - 67.36 \exp\left(\frac{[\text{heterodimer}] + 6.226}{7.0145}\right), \tag{4}$$

which can be used as a generalized equation to predict $J_{1.5}$ for acid–base particle formation at any (atmospheric relevant) conditions, given a calculated $\Delta G_{\text{heterodimer}}$ and temperature and concentration. Because $\Delta G_{\text{heterodimer}}$ requires significantly

less computational power to calculate than $J_{1.5}$, this trendline provides a method to quickly approximate $J_{1.5}$.

Since the heterodimer concentration is still affected by changes in temperature and concentration, Equation 4 is only able to approximate $J_{1.5}$ to within 10 orders of magnitude. This is because of the large range of temperatures and concentrations calculated in this study, where, in general, for concentrations less than $10^7 \, \text{cm}^{-3}$ and temperatures greater than 298 K, predicted $J_{1.5}$ are below the trendline. Similarly, for concentrations more than $10^7 \, \text{cm}^{-3}$ and temperatures greater than 298 K, predicted

$J_{1.5}$ are above the trendline, which can be seen in Figure 11b.

Though the 10 orders of magnitude uncertainty is large, Pierce and Adams (2009) have shown that 6 orders of magnitude uncertainty in new particle formation events in the atmosphere only contributed to a difference of 17% in modeled concentrations of cloud condensation nuclei (CCN) in the troposphere. Considering the simplicity of this calculation, this approach may improve estimates of global CCN in models that are limited by the computational expense of calculating $J_{1.5}$.

### 3.4    System-specific parameterization for weak bases using normalized heterodimer concentration ($\Phi$)

Here we attempt to reduce this uncertainty for nine salts of sa and further simplify the expression used to calculate $J_{1.5}$. We accomplish this by incorporating heterodimer concentration and monomer concentrations into a new independent variable, the normalized heterodimer concentration, $\Phi$:

$$\Phi = \frac{[\text{heterodimer}]}{\left(\frac{[\text{acid}][\text{base}]}{C_{\text{ref}}}\right)^{1/2}}, \tag{5}$$

When applied to ammonia, a simple monotonic relationship between $\Phi$ and $J_{1.5}$ becomes immediately apparent (Figure 12a). Here we observe that temperature affects the value of $\Phi$ minimally, and that the effects of temperature and concentration are incorporated in the dependent variable resulting in relatively minor data spread. Again, CLOUD $\Phi$ values were calculated for comparison, and CLOUD data are all predicted within 2 orders of magnitude of the best exponential fit to the data (fit equation available in the SI). The dispersion in $J_{1.5}$ remains constant over all conditions explored.

As a contrast to the sa–amm system, we also examined the behavior of the sa–gua salt, a strong-acid and strong-base combination. Figure 12b shows that a monotonic relationship does not apply for such systems. In fact, at each concentration, $J_{1.5}$ quickly reaches the kinetic limit and remains constant with temperature once monomer concentrations are above $10^7$ $\text{cm}^{-3}$. Gua is likely insensitive to changes in temperature because gua is a strong base and forms more stable growing clusters than those of ammonia. In addition, at higher concentrations than $10^7 \, \text{cm}^{-3}$, collisions are occurring so quickly that if the

cluster evaporates a monomer, another monomer is able to readily take its place. In this way, gua salt $J_{1.5}$ are largely dictated by monomer concentration rather than temperature.

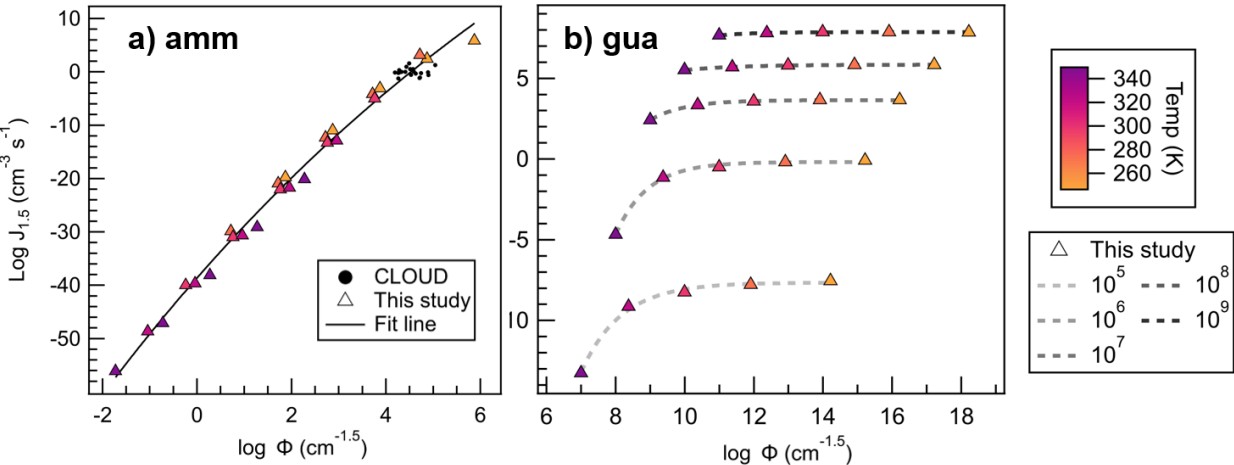

**Figure 12.** A) Amm and b) gua sa salts' $J_{1.5}$ plotted against $\Phi$, where triangles are colored according the temperature of that point's calculation. CLOUD data are shown as black dots, and their $\Phi$ values were calculated according to Equation 5. All trendlines used an exponential fit.

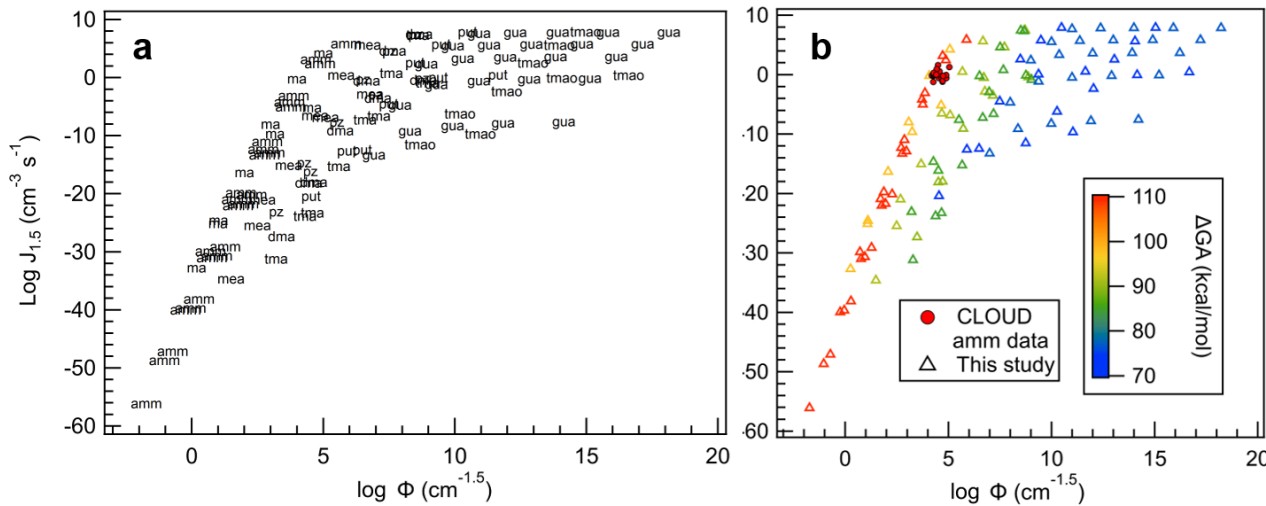

**Figure 13.** All sa salts plotted with a) base names as markers and b) markers colored according to their $\Delta$GA values. CLOUD observations are shown as filled circles.

When $\Phi$ is compared to $J_{1.5}$ for all bases (Figure 13a), we can immediately see that, in general, each base follows a unique trendline. Additionally, more bases follow the more monotonic behavior of sa–amm than sa–gua and increase in the data





dispersion follows increasing basicity. This is apparent when each of the base datapoints are colored according to their $\Delta$GA values (Figure 13b). In general, the larger $\Delta$GA values correspond to more linear, less dispersed relationships between $J_{1.5}$ and $\Phi$, and as $\Delta$GA decreases, $J_{1.5}$ begin to saturate and dispersion increases. This change in behavior seems occur most dramatically as $\Delta$GA decreases below 90 kcal/mol for the conditions shown here; however, it is likely for larger concentrations

or lower temperatures, even the weakest of bases will saturate. The fact that $\Delta$GA is directly linked to $J_{1.5}$ saturation highlights how acid and base strength are crucial to understanding cluster formation and growth into particles.

Here, $\Phi$ can be used to predict $J_{1.5}$ relatively accurately for specific bases, as demonstrated by the CLOUD $J_{1.7}$ observations. However, for bases with $\Delta$GA below approximately 90 kcal/mol, prediction becomes more uncertain as the kinetic limit becomes easier to reach. This $\Delta$GA cutoff of 90 kcal/mol means that the most abundant bases in the atmosphere, amm, ma,

dma, and tma, are not expected to saturate in this model under atmospheric conditions and thus their $J_{1.5}$ can be approximated relatively accurately using the results of this study. While this can only be used for experiments with acid and base monomer concentrations within 50% of each other over the concentrations and temperatures studied, this is a powerful predictive tool using only the term, $\Phi$, which only requires the calculation of one computational parameter, $\Delta G_{\mathrm{heterodimer}}$.

Because each base has its own correlation between $\Phi$ and $J_{1.5}$, the trendlines here cannot be generalized to bases that are not

described. For those bases not described here, Equation 4 should be used to approximate $J_{1.5}$ to within 10 orders of magnitude.

## 4   Conclusions

Here we have shown that heterodimer stability is largely predicted by the gas-phase acidity of the constituent acid and base across 27 acid–base pairs. In addition, we found that trends between heterodimer stability and physical properties such as volatility, dipole moment, and polarizability did not hold for the wide variety of bases studied here, despite a trend existing

for the smaller set of amm, ma, dma, and tma. We emphasize here the importance of studying a variety of bases with different structures and physical properties in order to make sure our understanding of salt NPF remains unbiased. We have also shown the relationship between $J_{1.5}$ and heterodimer stability and how it was affected by temperature and concentration. We show that deviations from the lognormal relationship were attributed to the remaining HBD available on the base molecule on the heterodimer. Then in order to devise a simple model to predict $J_{1.5}$, we calculated heterodimer concentration from our

heterodimer stability values. The effects of temperature and concentration on heterodimer concentration were much less than that of those on $\Delta G_{\mathrm{heterodimer}}$ but still were present, as shown by the 25 different calculations of sa–amm $J_{1.5}$. When compared to CLOUD experimental $J_{1.7}$ data, the sa–amm trendlines were able to predict $J_{1.5}$ within two orders of magnitude when the closest temperature trendline was used. We found that heterodimer concentration can be parameterized into a expression that can predict $J_{1.5}$. Because of this, the more difficult to calculate parameter of $J_{1.5}$ could be replaced by the more easily acquired

parameter of heterodimer stability. In addition, we have calculated a new parameter, the normalized heterodimer concentration, $\Phi$, which minimized the effects of temperature and concentration even more than that of heterodimer concentration. We found that $\Phi$ reduces the complexity of calculating $J_{1.5}$ by producing a single, monotonic trendline for sa–amm, instead of 10 as it was for our calculations using heterodimer concentration as the independent variable. The ability of $\Phi$ to accurately predict $J_{1.5}$



applies to sa salts of weaker bases, as stronger bases quickly saturated to reach the kinetic limit. This behavior was exhibited more strongly for salts that had a $\Delta$GA value smaller than 90 kcal/mol.

In addition, we have presented a facile way of predicting $J_{1.5}$ to within 10 orders of magnitude for salts of sa using a generalized parameterization (Equation 4). We also present a method to more accurately predict $J_{1.5}$ using the new parameter $\Phi$

for the nine sa salts studied here. It is important to note that, due to computational restrictions, all particle formation simulations are performed for two-component neutral clusters with an absence of relative humidity. Thus theoretical results might vary compared to measured particle formation under atmospheric or laboratory conditions. Water enhancement of NPF is known to be greater with more available hydrogen bonding sites as shown in Yang et al. (2018), which may enhance the deviation from the lognormal relationship that was attributed to remaining HBD on the heterodimer. The enhancing effect of ions on the

NPF rate can be several orders of magnitude for systems where small neutral clusters are unstable (e.g., ammonium salts in this study), but is negligible with more stable clusters, like a strong acid and base pair (Myllys et al., 2019b). In addition, when more than two components are present at the same time in the atmosphere or even as a contaminant in laboratory, NPF can be largely enhanced due to synergistic effects (Glasoe et al., 2015; Jen et al., 2014; Yu et al., 2012; Temelso et al., 2018; Myllys et al., 2019a). It is infeasible to explicitly study of all possible combinations of multi-component acid and base mixtures, but

perhaps in the future the synergy between different compounds and the role of water vapor could be estimated using some simple parameters such as GA values and number of hydrogen bonding sites.

*Author contributions.* NM performed the quantum chemical calculations and cluster formation simulations and initiated this study's design and supervised the data analysis process. SC curated, analyzed, investigated, and visualized cluster data. SC prepared the bulk of the manuscript with contributions from NM, JNS, and KB. JNS and KB acquired funding and supervised the project. All authors have read and

agreed to the published version of the manuscript.

*Competing interests.* The authors declare that they have no conflict of interests.

*Acknowledgements.* We thank the CSC-IT Center for Science in Espoo, Finland, for computational resources. NM thanks the Jenny and Antti Wihuri foundation for financial support. JS acknowledges funding from the U.S. National Science Foundation under grant no. CHE-1710580.



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



*Code and data availability.* The supporting information contains the following sections:

- – Monomer structures and properties

- – Methanesulfonic acid and nitric acid complexes

- – Acidity measures

5 – Base dipole moment and polarizability

- – Boundary conditions in particle formation simulations

- – Simulated particle formation rates

- – Hydrogen bonding in clusters

- – Predictive expressions of $J_{1.5}$ for ammonia