# Peer review of "A Predictive Model for Salt Nanoparticle Formation Using Heterodimer Stability Calculations"

_Atmospheric Chemistry and Physics, 2021_

## Author Comment (AC1)

UNIVERSITY OF CALIFORNIA, IRVINE

BERKELEY • DAVIS • IRVINE • LOS ANGELES • MERCED • RIVERSIDE • SAN DIEGO • SAN FRANCISCO    SANTA BARBARA • SANTA CRUZ

[Figure]

Department of Chemistry

1102 Natural Sciences 2
University of California, Irvine
Irvine, CA  92697-2025

June 29, 2021

Dear Editor:

We highly appreciate the constructive comments from the reviewers, and we have addressed the comments in the revised paper. We hope that the following responses are satisfactory and that the paper can be accepted for publication in Atmospheric Chemistry and Physics without further delay. The reviewers' comments have been reproduced in blue text below, followed by our point-by-point replies.

Reviewer comments:

**Reviewer: 1**

S. Chee and co-workers have used computational methods to study the stabilities and formation rates of acid-base clusters relevant to atmospheric new-particle formation. The study has two main parts. First, the authors investigate whether the formation free energy ("stability") of a "heterodimer" (a cluster of one acid molecule and one base molecule) can be predicted based on various single-molecule properties, including both experimental and easily computed parameters. Second, the authors then study how well this heterodimer stability correlates with particle formation rates computed with a cluster population dynamic model using quantum chemistry data (at the same level) as input. The study is interesting, well carried out, and useful to the atmospheric aerosol community. In addition to the main results, the study also has some very useful discussion connecting various molecular and/or cluster properties or property trends to the actual structures. I especially liked figures 3 and 6, and the associated discussion. I thus recommend publication in ACP subject to some fairly minor revisions.

**Author reply:**

We highly appreciate these valuable comments and suggestions.

Suggestions for revision:

-Please define what is meant by a "weak salt" e.g. in the abstract (or rephrase this).

**Author reply:**

We have clarified "weak salt" as a salt having a difference in gas-phase acidity greater than 95 kcal/mol, like ammonium sulfate and methylaminium sulfate in both the abstract and the end of the introduction.

-When the authors discuss "gas phase acidity", as well as pKa, they mean both the acidities of the studied acids, and the acidities of of the conjugate acids of the bases (which are in turn measures of the "basicities" of the parent bases). Or in other words, when they discuss for example the "pKa of methylamine", they do NOT literally mean the negative base-10 logarithm of the equilibrium constant for the reaction $CH_3NH_2 + H_2O \Longleftrightarrow CH_2NH_2^- + H_3O^+$ (that would be the technical literal interpretation of the phrase), but instead that of the reaction $CH_3NH_3^+ + H_2O \Longleftrightarrow CH_3NH_2 + H_3O^+$. This is obvious to most chemists, but not necessarily to all

physicists. I suggest the authors explicitly explain/define this notation early on in the manuscript (now this is implicitly mentioned only on page 8).

**Author reply:**

Indeed, this might be confusing. We have added clarification to Table 2 as: "Acidity of an acid \ce{HA} or bases' conjugate acid \ce{BH+}". In addition, we have added the following clause to the beginning of the gas-phase vs. aqueous-phase acidity section: "wherein we define acidity of a base to be the acidity of the conjugate acid (i.e., the gas-phase or aqueous-phase acidity of NH3 refers to the acidity of the conjugate acid, NH4+)."

-As noted above, the study really has two quite separate parts: the prediction of heterodimer stability on one hand, and the prediction of J1.5 based on that stability on the other hand. Also while the first is done with three different acids, the second is apparently only done for H2SO4 - base clusters. The abstract does not really make this clear, and also the two topics are presented in a somewhat counterintuitive order. I would suggest some rephrasing and rewording of the abstract to make the content and structure of the study more clear.

**Author reply:**

Good catch. We have reordered the abstract to reflect the correct paper order.

-In section 2.1, the authors say "In order to simulate cluster formation and growth, one must calculate accurate structures and thermochemical properties of neutral sa–base clusters up to the cluster size of four sa and four base molecules". As the authors must know, the appropriate "box size limit" depends both on the system (which acid and which base) and on the conditions (concentrations and temperature). 4,4 is not some universally valid constant. See for example Besel et al, https://pubs.acs.org/doi/abs/10.1021/acs.jpca.0c03984, for some discussion, and on how to check if the box size is suitable (by comparing the evaporation and collision rates of the most stable of the largest included clusters). The authors should add a few sentences of discussion on this, and also check if the 4,4 box is appropriate for all the studied cases. I would expect based on the study quoted above that especially for H2SO4:NH3, at the lowest concentrations (1E5 per cm3) and highest temperatures (348 K!), the box size may be (way) too small. A caveat on this should be added, and it would be good if the authors could indicate also in their figures which of their formation rates may be overestimates due to box size effects. (This should mainly affect rates that are anyway quite low, so this is not a huge problem, but it ought to be properly documented.)

**Author reply:**

We agree that especially in the case of ammonia, the box size is likely to be too small and that leads overestimated particle formation values. We added the following clarification to the Section 2.2: "Additionally, the simulation box size of 4 acid and 4 base molecules might be too small (i.e., critical cluster is outside of a box) at high temperatures and low concentrations. This leads overestimated particle formation rates as discussed in \cite{besel20}, where the effect of simulation settings was studied in the case of ammonia and sulfuric acid nucleation."

-The first sentence of section 2.2 should slightly be amended to reflect the above: yes the methods can in principle be used to simulate "any conditions", but for weakly bound clusters, low concentrations and high temperatures, the set of included clusters may need to be expanded if accurate rates are desired.

**Author reply:**

This is true. Changed from "at any conditions" to "at conditions where particle formation rates are not experimentally measurable".

-Give a few more details on the conformational sampling please: the Kubecka et al 2019 study contains quite a few different options and possible parameter (e.g. cut-off energy) values. Also, did the authors use e.g. quasi-harmonic corrections?

**Author reply:**

We provide more information about the sampling settings and computational details.

Added to Section 2.1 to give details about sampling procedure: "Briefly, to create the initial cluster structures, we used 3000 random guesses and 100 exploration loops, with a scout limit of 4 in the ABCluster program, and for each building block combination we saved 300 of the lowest energy structures that were subsequently optimized by the tight-binding method GFN2-xTB with a very tight optimization criteria~\citep{ab-rigid,abcluster,xtb2}. Based on the electronic energies, radius of gyration, and dipole moments, we separated different conformers, which were then optimized using the \ce{\omega}B97X-D/6-31+G* level of theory. Based on the obtained electronic energies, we selected structures with a maximum of $N$ kcal/mol from the lowest electronic energy (where $N$ is the number of molecules in the cluster)."

Added to Section 2.1 to specify usage of RRHO: "using rigid rotor--harmonic oscillator approximation"

-Also explain a bit more about what ACDC does, e.g. the fact that collision rates correspond to hard-spheres, and then evaporation rates are computed from quantum chemical free energies using detailed balance.

**Author reply:**

We have added to Section 2.2 more details: "The collision coefficients are computed from kinetic gas theory and the evaporation rates from quantum chemical Gibbs free energies assuming detailed balance."

-I suggest using capital letters for the abbreviations (so tma => TMA etc); e.g. "put" is easily confused with the corresponding verb when written in lower case.

**Author reply:**

We thank the reviewer for the suggestion. Changes have been made.

-Page 11, "volatility of the constituent acid and base plays a relatively minor role in heterodimer stability". This is true as stated, but maybe to avoid misunderstandings you could note in this discussion that e.g. the low volatility of H2SO4 is still a big part of why this molecule is so important for atmospheric new-particle formation. (This does not contradict anything the authors state, it's just a complementary fact.)

**Author reply:**

This is a good point. We aim to further discuss the effect of volatility in the future, especially in the context of particle's acid-to-base ratios. We have also added a clause in the text after the comment on volatility's minor role in heterodimer stability: "However, volatility plays a key role in cluster and nanoparticle growth, wherein low volatility compounds in the atmosphere (i.e., \ce{H2SO4}) are still very important for understanding NPF."

-Please explain what is meant by "lognormal" in the context of Fig 8a.

**Author reply:**

Lognormal here refers to the fit of the trendline. We have removed the word from the discussion and rephrased the sentence as: "the fit curve shape remains the same as \ce{$J$_{4x4}} is shifted upwards with increasing starting concentrations until the kinetic limit is reached"

-W(1/2) in equation 2 should presumably be W(1,2). Also please explain this notation in a bit more detail. E.g. p(1,2) is the number concentration of what species? (Also presumably J is proportional to concentration, not the other way around!

**Author reply:**

Thanks for spotting the typo. We have explained p(1,2) in more detail in the text and rearranged the sentence to make sure J is proportional to concentration: "$J$ is directly proportional to heterodimer concentrations ($p(1,2)$, which is related to the starting concentrations of acid and base), whereas temperature contributes from within the exponential expression, which matches the behavior seen in Figure 8a"

-Please confirm that the delta-Gs are calculated from partition functions at each temperature (and not from equation 1 assuming temperature-idependent delta-H and delta-S, which would introdue a completely unnecessary extra error source). Also you could note if the calculations are based on lowest free-energy minima (at 298 K, or checked at each temperature?), or if multiple minima are included.

**Author reply:**

We have only used the global minimum energy structures found at 298 K and at other temperatures the Gibbs free energies are calculated from enthalpy and entropy values calculated for that structure at 298 K. Indeed, this might introduce an additional error, and this is now commented in Section 2.2 as: "It should be noted that the cluster sampling procedure were performed at 298~K and those structures and thermodynamic data (H and S) have been used in simulations at all temperatures. Thus, at lower or higher temperatures, slightly different global minimum structures might exist."

-"deltaGheterodimer predicts theoretical J 1.5 well at cold temperatures, but additional factors become more prominent at warmer temperatures". Isn't this just reflecting the fact that especially for the stronger bases, evaporation of clusters larger than the heterodimer are negligible for lower temperatures, so J is then determined by the heterodimer evaporation rate? While, for higher temperatures also evaporation rates of larger clusters (which correlate with, but are not directly determined by, the heterodimer delta-G) start to matter?

**Author reply:**

Yes, this is correct. At high temperatures larger clusters evaporate as well which affect to the particle formation efficiency (and this is consistent with reviewer's comment that the 4x4 box size might not be large enough for weak bases at high temperatures).

-Is Figure 11 only for SA:AMM, or for all bases with SA? (Based on the text probably the latter, but I'm not 100% sure.) Please note this in the caption.

**Author reply:**

It is for all sulfuric acid salts which is now specified in the caption.

-Comparing equations 3 and 5, the normalised heterodimer concentration seems to be simply ([acid][base]/c_ref)^0.5 x exp(-dG/RT). I.e. the same as the mass balance equation but with a square root around the prefactor. This could perhaps be noted. Also, to me this seems to be maybe be somehow related to the concept of saturation ratio (perhaps it depicts the saturation of the monomers with respect to the heterodimer)? If the authors have any insight into such a connection, please feel free to speculate :-).

**Author reply:**

Indeed, this is true. We acknowledge this in Section 3.4 as "It can be noted that Equation \ref{phi} is same as Equation \ref{hetdim} but with a square root around the concentration term."

-In the conclusion, the authors write "The effects of temperature and concentration on heterodimer concentration were much less than that of those on deltaGheterodimer". I was under the impression that their delta-G was defined for c_ref = 1 atm, i.e. the actual value should not depend on the monomer concentrations. Perhaps they need to reformulate this sentence?

**Author reply:**

Yes, this is confusing sentence as deltaG has only linear dependency on the temperature as Equation 1 shows. We modified the sentence as: "Then in order to devise a simple model to predict \ce{$J$_{4x4}}, heterodimer stability values were used to calculate heterodimer concentrations. Indeed, the relationship between heterodimer concentration and \ce{$J$_{4x4}} varied much less with changes to temperature and concentration compared to the relationship between \ce{$\Delta G$_{heterodimer}} and \ce{$J$_{4x4}}"

In addition, we modified section 3.3 to reflect the same conclusion: "the \ce{$\Delta G$_{heterodimer}} vs. \ce{$J$_{4x4}} relation varies strongly with temperature and concentration"

-As the authors themselves note, the proposed approach for using heterodimer stability to predict new-particle formation rates is valid only for acid-base clusters, and for conditions where the acid and base concentrations are fairly similar. In the literature, the stabilities of heterodimer of type (X)(H2SO4) is very often used to (in my opinion incorrectly) argue for e.g. "nucleation enhancement" by a great multitude of (usually non-basic oxidised organic) species X. For example, dicarboxylic acids, such as oxalic acid mentioned by the authors, often bind quite strongly to one H2SO4 molecule - but are incapable of promoting the addition of further H2SO4 to the cluster. In such cases, "heterodimer stability" is clearly NOT a sufficient or good indicator of particle formation efficiency. To avoid other researchers misrepresenting their results, I strongly recommend that the authors explicitly state that the proposed relation between heterodimer stability and new-particle formation holds ONLY for acid-base clusters, and NOT for the general case of H2SO4 clustered with an arbitrary other (non-basic) molecule.

**Author reply:**

This is a very important point!

Added to the end of Section 3.2.1: "It is worth of mentioning that our model for using heterodimer stability to predict particle formation rates is valid only for acid--base clusters, not for organic acid--inorganic acid clusters. Thus, for instance a formation free energy value of oxalic acid--sulfuric acid heterodimer cannot be used to predict particle formation efficiency using any formula presented in this paper."

And to the Section 4: "It is important to emphasize that all of these predictions of \ce{$J$_{4x4}} based on heterodimer stability is only possible for heterodimers made up of one acid and one base molecule, and not any other combination of molecules wherein the word heterodimer may apply."

---

## Author Comment (AC2)

Department of Chemistry

1102 Natural Sciences 2
University of California, Irvine
Irvine, CA  92697-2025

June 29, 2021

Dear Editor:

We highly appreciate the constructive comments from the reviewers, and we have addressed the comments in the revised paper. We hope that the following responses are satisfactory and that the paper can be accepted for publication in Atmospheric Chemistry and Physics without further delay. The reviewers' comments have been reproduced in blue text below, followed by our point-by-point replies.

Reviewer comments:

**Reviewer: 2**

Chee et al. present a detailed review on how various atmospheric bases react with sulfuric acid/methane sulfonic acid/nitric acid to nucleate particles. They use the previous published computational chemistry data on cluster formation energies for these acid-base systems to draw several conclusions on what type of molecule is needed to nucleate with atmospheric acids. They found gas-phase acidity to be the best indicator of how stable the heterodimer was and thus nucleation rates.  It is quite satisfying to read a paper that shows that vapor pressure is not a good predictor of (acid-base) nucleation. This paper fits ACP well and is a great article on how to think about acid-base nucleation in the atmosphere. There a few points the authors should address before this manuscript should be accepted for publication.

**Author reply:**

We are grateful of these relevant comments and suggestions.

Specific comments:

Why did the authors decide on J1.5 nm? This size makes it difficult to compare with published observations of J1.7 nm or J1.0 nm. Along this same line, the authors compare their calculated J1.5 nm to CLOUD's J1.7 nm. The authors should comment on how the smaller diameter size will impact that comparison.

**Author reply:**

The value J1.5nm corresponds a formation rate of clusters larger than 4 acid and 4 base molecules, for which the diameter is around 1.5 nm. In ACDC simulations we define a simulation box size (4 acid and 4 base) and the growing out criteria (stable clusters larger than 4 acid 4 base clusters, specific criteria for each system is given in SI). Thus, in contrast to measurements, simulations do not have an exact diameter which are detected, but a combination of molecules in cluster, and we approximated the size of clusters larger than 4 acid and 4 base molecules to be 1.5 nm, which might be underestimated for large bases such as putrescine and overestimated for small bases such as ammonia. We have changed J1.5 to J4x4 to reduce overanalysis of the data.

The CLOUD data does not span enough orders of magnitude to merit the statement that their model for J1.5 nm to be accurate to measurements within 2 orders of magnitude. It would be helpful if the authors could either compare to more observations or re-evaluated their conclusion from comparing to CLOUD data. The authors mention that they can only compare to data where acid concentrations are approximately equal to base concentrations. Dr. Hanson at Augsburg College has published results where acid and base concentrations are approximately equal and has explored numerous bases.

**Author reply:**

We agree that more comparison with measurements would be useful. In future, we aim to further develop our model and include effect of ions and hydration which allows wider comparison with experimental results.

This may be outside the scope of the study but several papers have been published recently examining organic acid+base nucleation: Chen et al., 2017; Kumar et al., 2019 and other papers from Hansen and Francisco. If possible, it would be helpful to put their energy calculations into context with the results shown here.

**Author reply:**

We agree that organic acid+base nucleation is worth of studying in the future.

The authors present their J1.5 model as a function of [heterodimer] as simple and relatively accurate. 10 orders of magnitude is quite large. Though Pierce and Adams (2009) show 6 orders of magnitude may not be a big deal in predicting particle concentrations, what about 10 orders of magnitude? Also the presented model is based on their calculated J1.5 from their computational chemistry results. The comparison with CLOUD data does not provide a good indication how accurate their J1.5nm is to observed J1.5 (which would include water). It would be helpful if the authors could provide a short discussion on uncertainties in their J1.5 calculation so the reader knows how well equation 4 does in predicting observed J1.5.

**Author reply:**

We agree that 10 orders of magnitude is quite large and have removed the recommendation for the equation to be used in global models. We added a brief discussion on the effect of water: "It is important to keep in mind that this model was calculated in the absence of relative humidity, which may enhance \ce{$J$_{4x4}} rates for those acid--base pairs with many free HBD~\citep{directlink}."

From the abstract, I was expecting the normalized heterodimer concentration to estimate nucleation rates. However, it seems this is not true as it really only works for ammonia and methylamine. Basically all the other bases presented here fall off the linear curve presented in figure 12 and 13. In addition, the authors provide quite a few caveats to using this parameter to estimate J, like acid and base concentrations need to be approximately equal. In the atmosphere, ammonia is almost always higher in concentration than sulfuric acid. (The other bases are so poorly measured around the world that it is hard to say how their concentration varies.) In which case, I am not quite sure the purpose of this normalized concentration parameter? If the authors are very committed to keeping this parameter, it would be helpful then to define what a weak salt is in the abstract. Also it would be very helpful to include an equation showing how to calculate J from $\Phi$.

**Author reply:**

We agree that the applicability of the parameter is limited with its current caveats as mentioned. We believe this to be a good starting point for future studies that can explore more atmospherically relevant conditions to provide better estimations of J from heterodimer stability (or some permutation thereof). A weak salt has been defined in the abstract as a salt with a $\Delta GA > 95$ kcal/mol, which encapsulates both ammonium sulfate (109 kcal/mol) and methylaminium sulfate (97 kcal/mol).

From the SI, the CLOUD that is being used also includes ion nucleation experiments. How are ion nucleation reactions taken into account with the heterodimer energies used in this study? Wouldn't ion cluster formation energies be drastically different than their electrically neutral counterpart?

**Author reply:**

That is a good point. We purposely did not take into account ion nucleation reactions in order to see if the nucleation experiment was still able to be described by the model we have proposed, and only one experiment has a large ionization rate. We aim to explore ion cluster formation energies compared to neutral clusters, as well as their effect on J rates in the future.

Technical Comments:

Page 2 line 20: Heterodimer stability reminds me of papers from (Kürten et al., 2014; Jen et al., 2014). Worth referencing them as they measured sulfuric acid heterodimer concentrations for the abundant atmospheric bases and concluded that how the dimer forms (and if they evaporate) is an important controlling factor for nucleation.'

**Author reply:**

Thanks for the added citations.

Page 4 line 10: how do the authors know 4 acids and 4 bases is 1.5 nm? Is this geometric diameter?

**Author reply:**

We have approximated the stable clusters larger than 4 acid and 4 base molecules to be 1.5 nm. The geometric diameters of 4 acid 4 base clusters are given in SI.

Page 4 line 30: and collected from

**Author reply:**

Thanks for spotting this.

Page 17 line 20: The normalized heterodimer concentration has units of cm^-1.5. Is this correct? I thought it would have units of cm^0.5.

**Author reply:**

The normalized heterodimer concentration is calculated from

$[heterodimer]/([acid][base]/C_{ref})^{1/2}$,

or,

$cm^{-3} / (cm^{-3} * cm^{-3} / cm^{-3})^{1/2}$,

which can be simplified to:

$cm^{-3} / cm^{-1.5} = cm^{-1.5}$

Figure 12: what are the dashed lines? Concentrations of what? Also it's really difficult to tell the difference between the different shades of gray.

**Author reply:**

The dashed lines are the monomer concentration of the acid and base in molec $cm^{-3}$. The lines have been changed to different dashed line types to help differentiate them.

References used in this review:

Chen, J., Jiang, S., Liu, Y.-R., Huang, T., Wang, C.-Y., Miao, S.-K., Wang, Z.-Q., Zhang, Y., and Huang, W.: Interaction of oxalic acid with dimethylamine and its atmospheric implications, RSC Adv., 7, 6374–6388, https://doi.org/10.1039/C6RA27945G, 2017.

Jen, C. N., McMurry, P. H., and Hanson, D. R.: Stabilization of sulfuric acid dimers by ammonia, methylamine, dimethylamine, and trimethylamine, J. Geophys. Res. Atmospheres, 119, 2014JD021592, https://doi.org/10.1002/2014JD021592, 2014.

Kumar, M., Burrell, E., Hansen, J. C., and Francisco, J. S.: Molecular insights into organic particulate formation, Commun. Chem., 2, 1–10, https://doi.org/10.1038/s42004-019-0183-7, 2019.

Kürten, A., Jokinen, T., Simon, M., Sipilä, M., Sarnela, N., Junninen, H., Adamov, A., Almeida, J., Amorim, A., Bianchi, F., Breitenlechner, M., Dommen, J., Donahue, N. M., Duplissy, J., Ehrhart, S., Flagan, R. C., Franchin, A., Hakala, J., Hansel, A., Heinritzi, M., Hutterli, M., Kangasluoma, J., Kirkby, J., Laaksonen, A., Lehtipalo, K., Leiminger, M., Makhmutov, V., Mathot, S., Onnela, A., Petäjä, T., Praplan, A. P., Riccobono, F., Rissanen, M. P., Rondo, L., Schobesberger, S., Seinfeld, J. H., Steiner, G., Tomé, A., Tröstl, J., Winkler, P. M., Williamson, C., Wimmer, D., Ye, P., Baltensperger, U., Carslaw, K. S., Kulmala, M., Worsnop, D. R., and Curtius, J.: Neutral molecular cluster formation of sulfuric acid–dimethylamine observed in real time under atmospheric conditions, Proc. Natl. Acad. Sci., https://doi.org/10.1073/pnas.1404853111, 2014.